# Development and Field-Testing of Proposed Food-Based Dietary Guideline Messages and Images amongst Consumers in Tanzania

**DOI:** 10.3390/nu14132705

**Published:** 2022-06-29

**Authors:** Lisanne M. Du Plessis, Nophiwe Job, Angela Coetzee, Shân Fischer, Mercy P. Chikoko, Maya Adam, Penelope Love

**Affiliations:** 1Division of Human Nutrition, Department of Global Health, Faculty of Medicine and Health Sciences, Stellenbosch University, Cape Town 7500, South Africa; 2Digital Medic South Africa, Stanford Center for Health Education, Cape Town 8000, South Africa; njob@stanford.edu (N.J.); fischers@stanford.edu (S.F.); madam@stanford.edu (M.A.); 3Sustainability Institute, School for Public Leadership, Stellenbosch University, Stellenbosch 7600, South Africa; angela.coetzee@me.com; 4Food and Agriculture Organisation, Sub-Regional Office for Southern Africa, Harare 3730, Zimbabwe; mercy.chikoko@fao.org; 5Stanford Center for Health Education, Department of Pediatrics, Stanford School of Medicine, Stanford, CA 94305, USA; 6Institute for Physical Activity and Nutrition (IPAN), School of Exercise and Nutrition Sciences (SENS), Deakin University, Geelong, VIC 3216, Australia

**Keywords:** food-based dietary guidelines, messages, images, pre-testing, Tanzania, nutrition education

## Abstract

In this paper we report on the development and field-testing of proposed food-based dietary guideline (FBDG) messages among Tanzanian consumers. The messages were tested for cultural appropriateness, consumer understanding, acceptability, and feasibility. In addition, comprehension of the messages was assessed using culturally representative images for low literacy audiences. Focus group discussions were used as method for data collection. Results indicate that the core meaning of the proposed FBDG messages and images were understood and acceptable to the general population. However, participants felt that nutrition education would be required for improved comprehension. Feasibility was affected by some cultural differences, lack of nutrition knowledge, time constraints, and poverty. Suggestions were made for some rewording of certain messages and editing of certain images. It is recommended that the field-tested messages and images, incorporating the suggested changes, should be adopted. Once adopted, the FBDGs can be used to inform and engage various stakeholders, including parents, caregivers, healthcare providers and educators on appropriate nutritional practices for children and adults. They can also be used to guide implementation of relevant policies and programmes to contribute towards the achievement of sustainable healthy diets and healthy dietary patterns.

## 1. Introduction

The United Nations Food and Agriculture Organization (FAO) together with the World Health Organization (WHO) developed the concept of food-based dietary guidelines (FBDGs) in 1995 [1]. Country-specific FBDGs are evidence-based recommendations—simple advisory statements, that express the principles of nutrition education mostly in terms of foods, considering customary dietary patterns, ecological setting, socio-economic and cultural factors, and the biological and physical environment in which the population lives. They are intended to guide the broad public to consume a healthy diet which is both health promoting and protective against the development of malnutrition and non-communicable diseases (NCDs) [1,2].

FBDGs include messages and visual illustrations (images) to help consumers to implement them as part of a healthy lifestyle. It translates nutrient standards and recommendations (dietary goals and guidelines) into simple, practical advice on the types, and sometimes quantities, of various foods needed for healthy dietary patterns [3]. The main purpose of the messages and images is therefore to assist the consumer in choosing a diet adequate in nutrients that is protective of malnutrition and NCDs. FBDGs can be accompanied by a food guide, which is a graphic representation or image of some or all FBDGs, either as one single combined illustration or a series of images that indicates the foods and food groups that should be consumed as part of a healthy diet [4]. One way FAO supports member countries is to assist in the development, revision, and implementation of FBDGs and food guides, aligned with the current evidence base. Periodically, FAO conducts reviews on progress in FBDG development and use [1,5].

Close to 100 countries around the globe have developed FBDGs; however, there are currently only nine countries in Africa with finalised dietary guidelines. These countries include Benin, Kenya, Namibia, Nigeria, Seychelles, Sierra Leone, South Africa, Zambia and Ethiopia. With the increasing burden of malnutrition across the African continent, many countries are currently developing their first set of dietary guidelines, and FAO are encouraging more countries to follow suit [2].

In early 2019, the FAO put out a call for proposals to researchers for the development and field-testing of proposed FBDG messages and a food guide for, among others, Tanzania. This project formed part of the FAO’s larger objectives to improve diets and nutrition in Southern Africa by developing and testing FBDGs and food guides for various countries and ultimately to contribute to eradicating hunger, malnutrition, and food insecurity.

The Republic of Tanzania is the largest country in East Africa and includes the Mainland and Zanzibar. The country is divided into 21 regions on the Mainland and three on the island (Figure 1). The main local language is Kiswahili. The literacy level is 78%. About 80% of Tanzanians are subsistence farmers or fisherman. Agriculture includes coffee, sisal, tea, cotton, and cattle. The country exports gold, coffee, cashew nuts, manufactured good and cotton [6].

Since the early 2000s, Tanzania has experienced impressive economic growth, strong resilience to external fiscal shocks and a decline in basic needs poverty because of its vast resources, sociopolitical stability and economic reforms [7,8,9]. In 2018, the poverty rate declined from 34.4% to 26.4%; however, this reduction was surpassed by the rate of population growth, which resulted in an increase in the absolute number of poor people in Tanzania. Therefore, about 14 million people lived below the national poverty line of 49,320 Tanzanian Shilling per adult equivalent per month and about 26 million (about 49% of the population) lived below the USD 1.90 per person per day international poverty line in 2018 [9]. There are large gaps and disparities in the poverty distribution among geographic regions; however, poverty seems to be most concentrated in the Western and Lake zones of the country and lowest in the Eastern Zones [9]. Promoting healthy diets in low-income and middle-income countries (LMICs) can reduce social inequality in diet between the poor and the rich, especially when it targets disadvantaged population groups and because of both short-term and long-term economic benefits to households due to better health and educational outcomes [10].

Similarly to other LMICs, Tanzania suffers a triple burden of malnutrition (i.e., under-nutrition, micronutrient deficiencies and over-nutrition). According to the Tanzania Demographic and Health Survey (TDHS) 2010 [11] and 2015 [12], respectively, while stunting among young children has notably declined from 44% in 2010 to 34.5% in 2015 and further to 31.8% in 2018, figures remain high due to factors such as underweight among mothers, low birth weight, and low dietary diversity [13]. The National Nutrition Study conducted by the Tanzania Food and Nutrition Center in 2018 indicated that 7.3% of women of reproductive age (WRA) were underweight (body mass index (BMI) < 18.5 kg/m^2^) and 31.7% overweight (BMI > 25 kg/m^2^) [13]. World Bank data from 2019 estimated that NCDs such as cardiovascular diseases, cancers, diabetes, and chronic respiratory diseases may cause 74% of all deaths globally. The figure for Tanzania was 34% [14]. Evidence from the TDHS indicates an increasing prevalence of overweight and obesity; the proportion of women aged 15–49 years who are overweight and obese increased from 22% in 2010 [11] to 28% in 2015 [12] with uneven prevalence distribution between rural and urban areas, and the higher education and wealth quintiles being the most affected. Furthermore, the 2012 Tanzania STEPS study [15] reported the prevalence of hypertension to range from 27.1% to 32.2% and 28.6% to 31.5% in men and women, respectively, and the prevalence of elevated cholesterol to be 26% and obesity to be 35% among adults over 25 years.

An unhealthy diet is one of the most important risk factors that needs to be addressed to tackle the triple burden of malnutrition [16] and diet-related diseases [17,18] in LMICs such as Tanzania, yet there is a scarcity of national data on the patterns of dietary intake. In general, the majority of Tanzanians eat unbalanced diets high in staples (maize and rice) and small amounts of vegetables and fruit. In many areas, people have shifted from traditional diets to consumption of foods high in refined carbohydrates, sugar, fat and salt such as cereals and meat [15,19]. Tanzania’s changing food environment is characterized by declining total shares of household income spent on food [20] and increased access to non-staples, processed foods, edible fat and sugary beverages [9,19]. Available information indicates significant dietary gaps, including inadequate intakes of vitamin A, calcium, folate, zinc, iodine, and B vitamins leading to deficiencies, which are of economic and public health concern among all population groups in Tanzania [21]. In sub-Saharan African countries such as Tanzania, micronutrient intake has declined over the past 50 years, as shown by a reduced dietary micronutrient density index (average micronutrient density of the food supply based on 14 micronutrients, calcium, copper, iron, folate, magnesium, niacin, phosphorus, riboflavin, thiamin, vitamin A, vitamin B12, vitamin B6, vitamin C and zinc). Reasons for this include increased availability of grains (rice, maize, and wheat) and vegetable oils which have low micronutrient density, and decreased proportional availability of pulses, dairy products, meat, nuts and seeds, fruit, and vegetables [22]. Recent changes in dietary patterns in LMICs [23,24] as in Tanzania, indicate an important focus on improving diet quality for better health, prevention of diet-related diseases and reduction in the triple burden of malnutrition [24,25].

A critical consideration for nutrition education is that people eat foods and not individual nutrients. FBDGs must therefore be based on (i) the country’s nutrition-related public health issues, (ii) the availability, accessibility, and perception of the message content, and (iii) the acceptability to all populations taking into account their lifestyle, cultural eating habits and socio-economic circumstances [1]. Important considerations for Tanzania are the variation in food availability across regions, by season and in rural/urban settings. Another consideration is cultural and religious food preferences and taboos. The Tanzanian cuisine is a fusion of East Asian, Portuguese and Indian cooking and spices. Approximately 61% of the population is Christian, 35% Muslim, and 4% other religious groups. Zanzibar’s residents are 99% Muslim [26] and as such, certain food habits are distinct to specific areas of the country. Food systems in Tanzania are also not uniform; they vary according to location, socio-economic status, and food market infrastructure. Various agro-ecological zones have resulted in different dietary patterns. With the rise of supermarkets and fast-food outlets in big cities such as Dar es Salaam and an increasing demand for quality food products in sufficient quantities, a major issue is ensuring that consumers are properly guided on what foods to eat and where to access such foods at reasonable cost [13]. Developing and testing of FBDGs should therefore take cognizance of all these dimensions.

### Rationale for this Study

A research team from Stellenbosch University, under the leadership of the principal investigator (PI) and first author from the Division of Human Nutrition, was appointed in 2019 by the FAO Sub-Regional Office for Southern Africa (SFS) to test proposed FBDG messages and images for Tanzania. In this paper we report on the development and field-testing of the proposed FBDGs for cultural appropriateness, consumer understanding, acceptability, and feasibility among Tanzanian consumers. In addition, we assessed the comprehension of accompanying culturally representative visual illustrations (images) for low literacy audiences.

## 2. Materials and Methods

### 2.1. Development of the Draft FBDGs for Tanzania

The Tanzanian FBDG Technical Working Group (TWG) comprising of academics and government officials from various sectors, United Nations, and non-governmental organizations from the Republic of Tanzania (Mainland and Zanzibar), led by TFNC in collaboration with FAO, developed a set of 12 FBDG themes (Box 1) in early 2019 informed by public health nutrition and dietary data available for the country. The instruction to the research team was to refine these themes into preliminary FBDG messages. It was also expected that each message would have a visual illustration (image) that would inform one combined illustration (food guide). The research team was requested to support with the pre-testing of the messages and images.

Box 1Food-based dietary guideline themes suggested for Tanzania1. Eating a variety of foods from the following food groups: (a) vegetables, (b) fruits, (c) cereals/grains, starchy tubers or roots, (d) pulses, nuts, and seeds, (e) animal and animal products, milk and milk products, and (f) fruits and vegetables combined;2. Eating vegetables and fruits. Focusing on green leafy vegetables and orange and yellow coloured, non-citrus fruits and vegetables;3. Consumption of animal and animal products, milk and milk products;4. Consumption of pulses, nuts, and seeds daily;5. Limiting intakes of highly processed foods containing saturated fats and trans-fats, added sugar, and added salt;6. Infants and young children feeding (exclusive breastfeeding before 6 months and complimentary feeding after 6 months);7. Encouraging physical activity and exercise and maintaining a normal weight;8. Safe handling, preparation, cooking and storage of food including food handling to preserve nutrients (these could have 4 sub themes and illustrations);9. Practicing good food hygiene and sanitation such as washing hands with soap and clean water every time before preparing or eating food and at all critical times;10. Observing portion sizes for all food groups;11. Healthy eating for pregnant and lactating women;12. Providing nutritious meals and snacks to school-aged children.

From the 12 themes suggested by FAO and the TWG, the research team proposed preliminary FBDGs. These proposed preliminary FBDGs were based on FAO’s characteristics and criteria of FBDG messages [1,2,4], experience with testing FBDG in South Africa [27] and studying relevant scientific literature on the topic as well as country-specific literature for Tanzania.

During meetings in Zanzibar and Dar es Salaam (June 2019), the researchers, FAO delegates and the TGW members workshopped the preliminary FBDG messages. The research team suggested a life cycle approach for the development of images and refinement of the messages. This approach considers messages in all stages of human development, targeting different age groups of the population. The justification for a life cycle approach was based on the substantial, growing evidence of the impact of optimal nutrition during the first 1000 days of life (from conception to two years of age) on the rest of the human life span [28,29].

The research team also discussed the characteristic of sustainable diets in more detail and shared with the TWG some of the latest evidence on the “planetary healthy plate” [30] for possible inclusion in the field-testing process. Both suggestions (life cycle approach and planetary healthy plate) were agreed to. The messages were workshopped in much detail, and based on feedback from the TWG, changes were made to the text of the proposed draft FBDG messages. Messages were the same for both Mainland and Zanzibar, except for one distinct difference suggested for the vegetable message for Zanzibar and Mainland due to differences in cultural eating habits.

The final English messages were checked for translation by the TWG. These messages were deemed a final draft FBDG messages for field-testing (Table 1).

### 2.2. The Development of Images to Accompany FBDG Messages—Artwork Process

The research team aimed to develop images that were representative of the context. The graphic designer in the team led this process using mixed imagery, i.e., drawn images of persons and/or settings with photographs placed in the images for real and accurate representation of foods. To inform this process, during various meetings, the researchers asked pertinent questions to the FAO technical team and the TGW about the local food environment, foods that are culturally acceptable and available, and what should be included in the images. To create the artwork for the images of the FBDG messages, reference photographs were taken of food environments (e.g., food markets, retail stores), indigenous and locally consumed foods from all food groups, some tableware and a home environment. No persons were photographed.

Visits to local food markets and shops/supermarkets, accompanied by the FAO technical team and members of the TWG, were undertaken to obtain the required foods. Raw and cooked versions of foods, with branding removed if applicable, were photographed in a lightbox to minimise shadows and obtain a clear, well-lit photograph of the food, whole and in portioned amounts. These photographs were reworked and edited to create photographs which could be used in the images. Once the written guidelines were established, images were digitally drawn to illustrate them, placing the photographed food where necessary and appropriate.

### 2.3. Design and Setting

A qualitative study was conducted with focus group discussions as a data collection method. Qualitative research methods are most useful in understanding the viewpoints of individuals regarding sensitive and socially dependent concepts [31]. The study sub-sites (municipalities) were purposively selected in consultation with FAO and the TWG. The individuals represented in these groups collectively have extensive country-wide experience and attempted to include as much of the diversity of the population in the selection of the sites. The following sites were decided upon: (1) Dar es Salaam, (2) Mwanza and (3) Iringa Municipalities for the Mainland and (4) Kusini for Zanzibar. Dar es Salaam and Kusini represented urban sites; Mwanza and Iringa represented rural sites (Figure 1).

### 2.4. Study Population

Women were chosen as the study population as they are the primary family caregivers and generally make decisions regarding food choices and preparation. They may or may not receive support from partners/men, but they are the primary group who require empowerment to make the right decisions with regard to nutrition. Inclusion criteria were: women of 18 years and older, who speak and understand Kiswahili, and who were permanent residents of the chosen study sites. Exclusion criteria pertained to women who did not provide informed consent for participation, who had formal training in nutrition and those unable to participate in the study due to limited mental abilities or comprehension.

### 2.5. Sample Size

Sample size was determined by data saturation—a recognised qualitative research approach used where an investigator extracts data until sufficient information has been collected or until no new information can be obtained [32]. Three to four focus group discussions (FGDs) were planned in each of the four study sites (12–16 FGDs), consisting of 6–8 participants each, as per the recommendations by the FAO [1]. However, it was felt that 14 messages were too many to test in one FGD. The messages were therefore split so that 7 messages were tested per FGD with an anticipated 24–32 FGDs that could be conducted.

#### Sampling Strategy

Non-random purposive sampling was used to select participants for the FGDs, which is the appropriate sampling technique used for qualitative research [32]. Purposive sampling includes the process of intentionally selecting participants for the study based on certain characteristics [32].

In each selected study site, fieldworkers recruited potential individuals from households and various community-based settings. All of the identified participants were provided with information regarding the study. They were given a date and time of the FGDs and contacted by telephone or a visit from a fieldworker prior to the FGD, to promote attendance.

### 2.6. Training of Fieldworkers

As agreed between FAO and the TWG members, the TWG members (25 in total) acted as fieldworkers for this research study. The research team conducted FGD facilitation skills training, as well as feedback and reflection as part of pilot testing over a three-day period with the fieldworkers in August 2019 to embed theory and gain practical experience. Each fieldworker was supplied with a training manual to explain all the research processes. Training was presented in English, since all TWG members can speak, write, and understand English. The training sessions included an overview of the study’s aim and objectives as well as the inclusion and exclusion criteria and methodology. Fieldworkers were required to be familiar with all research documentation as well as the recruitment form which was used in the field to recruit participants.

The fieldworkers were thoroughly trained on the FGD guide. The FGD guide was translated into Kiswahili by trained professionals then back translated into English for quality control purposes. The role of the FGD guide was to provide the fieldworkers with direction while facilitating all discussions. Fieldworkers were also trained to take notes during FGDs regarding the dynamics of the group as well as body language and facial expressions of the participants. All these aspects were covered in the training session.

Since the final ethics clearance for the project was only received from the Zanzibar Health Research Institute by the time of pilot testing, and not from the Mainland Dar es Salaam office, all training and pilot pre-testing took place in Zanzibar on 23 August 2019.

### 2.7. Pilot Testing

Pilot field-testing was conducted as part of the fieldworker training and served two purposes. Firstly, it provided an opportunity for testing and validating content (messages and images), therefore gauging usefulness and relevance for the Tanzanian context. Secondly, it provided the fieldworkers with hands-on practical training on how to conduct FGDs using the FGD guide. Eleven “mock” FGDs were conducted. Fieldworkers worked in teams and transcribed the audio recordings. Four of the authors analysed the transcripts. This process assisted the research team to identify areas where further training was required, thereby consolidating learning and ensuring fieldworkers were prepared and confident to conduct the upcoming FGDs for the main study in their various regions.

From the pilot test findings, it was clear that the infant and young child feeding (IYCF) messages were already in use and well understood and communicated widely. In agreement with the TWG and FAO, a decision was therefore made to adopt the IYCF messages as detailed in Table 2 (messages 8–10) without needing further field testing.

In preparation for fieldwork, consent forms and questionnaires were translated into Kiswahili by trained professionals and translated back into English for quality control purposes.

### 2.8. Focus Group Discussions

Fieldwork took place in Zanzibar from 29 October 2019 to 3 November 2019 and on the Mainland from 25 November to 3 December 2019. Before commencement of the FGDs, all participants were requested to complete an informed consent form followed by a short socio-demographic questionnaire. The questionnaire consisted of information such as the participant’s date of birth, home language, highest level of education, employment status and role in the household pertaining to food. Fieldworkers guided the participants through the questionnaire by reading out all the questions and ensuring participants completed questionnaires correctly. The questionnaire was completed by a fieldworker in the case of a participant being illiterate.

All FGDs were approximately two hours in duration and were conducted at local libraries, town halls, school classrooms or suitable facilities at other community-based organisations. Facilities were contacted or visited in advance to arrange suitable dates and times for the FGDs. Venues were set up on the day of the discussion and refreshments provided. All FGDs were audio recorded, with consent from participants, for analysis purposes.

Each FGD was conducted by trained fieldworkers working in pairs. The FGD was guided by a discussion guide, which was adapted from similar studies (Appendix A) [27,33]. The discussion guide included the procedure that had to be followed, and questions to be asked and prompts that could be used to stimulate discussion on the proposed FBDGs and images. These included discussion points on previous exposure to similar messages, the mother’s/caregiver’s understanding and interpretation of messages and images, the perceived importance of the messages and images, as well as barriers and enablers to following of the messages. Participants were also asked how they would reword the message to make it more understandable to the general public. The guide was translated from English into Swahili by trained professionals. The document was back-translated into English for quality-control purposes. The same procedure was used for the translation of the socio-demographic questionnaires and consent forms.

The proposed FBDG messages and images were printed on flash cards and A3 size posters to assist discussions. FGDs began with fieldworkers welcoming the participants and explaining the main aim and expectations of the study. The fieldworkers read out one proposed message at a time, presenting a flash card of the proposed FBDG message, and facilitating discussions among the participants. The same format was used for the images. The proposed FBDG messages and images were numbered for field testing purposes, so that audio recordings could accurately reflect the message/image under discussion.

### 2.9. Data Analysis

After completion of FGDs, fieldworkers sent all audio files to the research team as detailed in the research protocol via “We Transfer”, an Internet-based computer file transfer service. The research team in turn forwarded the audio files to professional transcribers in Dar es Salaam, also via “We Transfer”, for verbatim transcription in Kiswahili followed by translation into English.

Three of the authors performed manual content analysis on the transcribed English translations. Each interview was assessed by at least two of the three authors. They independently inspected the data for common themes and coded the data, predominantly by deductive coding, after careful reading and re-reading of the text. Themes were organized according to broader clusters, based on the predetermined study objectives and the discussion guide. They discussed the analyses and reached consensus on the interpretation of the findings, as reported in the results section.

## 3. Results

### 3.1. Socio-Demographic Profile of Participants

Ten FGDs were conducted for Zanzibar (*n* = 58 participants) and 24 FGDs were conducted for Mainland (*n* = 266 participants), resulting in a total of 324 female participants. The FGDs were conducted with on average 6–8 (up to a maximum of 11) participants per group. The average age of participants was 35.01 ± 11.62 years, predominantly Kiswahili speaking (*n* = 322; 98%), and employed (*n* = 193; 59,6%) with 60,5% (*n* = 196) reaching at least a primary level of education. Most participants (*n* = 277; 85,5%) had children with an average age of 13.93 ± 10.48 years. Of the participants, 174 (53.7%) had children under the age of five years. Participants fulfilled various roles pertaining to food in the household, including deciding what food should be bought/used (*n* = 240; 74%), purchased (*n* = 252; 78%) and prepared (*n* = 302; 93%), growing food for own household use (*n* = 212; 65%) and selling produce for income (*n* = 158; 48,8%) (Table 2).

### 3.2. General Understanding of the Proposed FBDG Messages and Images, and Recommendations for Improvement

In general, the FGD participants in Zanzibar and Mainland understood the core meaning that the FBDG messages and images intended to convey. Participants reported that they had been exposed to similar messages mostly through radio, television, clinics/hospitals and schools. These were also the most suggested platforms for distribution of the final FBDG messages and images.

“*We need to use such foods so that we can put our body into a good health and our children to have strong minds and grow physically and mentally fit*.”(Participant 2 ZG2)

Participants reported that the messages and images complemented one another, and that the community would mostly be able to understand them as well. However, they did feel that education would be required for improved comprehension of the messages.

“*Yes, they will understand when they are exposed to education.*”(Participant 6 ZG2)

The messages aimed at school-aged children were not as familiar to participants and they expressed a need for more education on these messages specifically. During discussion, participants expressed appreciation for these messages, particularly making the link between a well-fed child being more receptive to education.

“*What I know is that in this message we are supposed to mobilize our children to eat food that will help build their health and also maintain their mental capability so as to manage well in their studies*.”(Participant 1 ZG3)

The messages and images on water, sanitation and hygiene (WASH) were very well known and it was clear that these had been widely communicated and distributed through government education initiatives.

Some suggestions were made to make messages and images more culturally appropriate, comprehensive, and understandable. Suggestions for cultural appropriateness were made in the Zanzibar context to cover women’s breasts in the breastfeeding image, cover girl’s legs in the school meal image, and separate men and women in the activity image. To make the messages and images more comprehensive and understandable, participants suggested adding specific food products or more examples of a specific food group (e.g., different fruits, vegetables and meats). A specific comment was made to label individual foods in certain images with text. However, when images were discussed, participants could clearly identify the foods, and since images are intended to stand independently, this suggestion was not incorporated in the final set of proposed FBGD messages and images.

### 3.3. Specific Understanding of the Proposed FBDG Messages and Images, and Recommendations for Improvement

In this section, each FBDG message is stated, and image portrayed, followed by a description of findings, specific suggestions for change and/or additions and deletions to the messages and images. Following on, the revised image and message is depicted, as relevant, Figure 2, Figure 3, Figure 4, Figure 5, Figure 6, Figure 7, Figure 8, Figure 9, Figure 10, Figure 11, Figure 12, Figure 13, Figure 14, Figure 15, Figure 16, Figure 17, Figure 18 and Figure 19 (copyright permission has been obtained from FAO and the artist (@Fao/Shân Fisch-er)). 

The message and image elicited good understanding from participants. The image supported understanding of the word “variety”, including that the food is “nutritious”, “healthy”, “balanced “/” mixed foods” and “good food”. In some instances, participants responded that not all members of the public will be able to understand and implement the message and image, due to poverty and the resultant inability to purchase some products.

*“Due to high levels of poverty in the villages, some members of the public will not understand this picture…”* (Participant 6 MM2)

Suggestions were made that the fluid in the image should be changed from juice to tea. The researchers suggested that water should be depicted in the glass to correspond with and enhance message and image 13. Since green leavy vegetables are commonly consumed, the green beans were removed and replaced with spinach. A recommendation was made from Mainland participants to replace the rice on the plate with *ugali* (stiff, cooked porridge) since *ugali* is considered the most consumed staple food in the country. Zanzibar participants, however, could relate to the rice on the plate. In addition, Zanzibar also favoured small fish (sardines) to the boiled egg.

The message and image were well understood, and no changes were suggested. Some discussion about beans causing ulcers occurred. This misinformation was apparently received as advice from doctors at hospitals and may be linked to the suggestion to avoid these foods in the case of irritable bowel syndrome (IBS). Since FBDG are aimed at the healthy population, this message was not altered to incorporate this specialised dietary advice.

A review of the term “nuts” in the Kiswahili message was suggested since there seemed to be ambiguity around the term. Some participants interpreted it as cashew nuts, and the image also depicted it. Most participants deemed cashew nuts as difficult to access or expensive. The image of cashew nuts was therefore removed, and since there is a wide variety of different pulses available in the country, more varieties were added to the revised image.

The message and image were well understood, and reference was made to “body building foods”. Animal source foodsi n Zanzibar mainly comprise seafood, whereas on the Mainland it is freshwater fish from the lakes and other animals. Participants from the Mainland could therefore not relate to the regular consumption of seafood, specifically octopus. A recommendation was made by Mainland FGDs to add red meat to the image. The image was changed to portray a wider variety of products (freshwater fish, octopus, red meat on a skewer as well as cooked chicken).

Indigenous foods, such as local insects, were recommended in a few instances in the Mainland FGDs. However, it is important to note that insects are generally region specific and would therefore require such consideration when included in education.

This message and the image were well received but elicited a lot of debate. The wording of “handfuls” was not well understood. Other methods of showing portion size were therefore suggested, such as “spoonfuls” made by Mainland FGDs or “bowls”. It was also suggested that frequency (“three times a day” and “included in three meals a day”) or listing the different types of vegetables by name.

There was a recommendation from Zanzibar to include okra in the image. Mainland suggested that green leafy vegetables, such as sweet potato leaves, cassava leaves and spinach should be included in the image as well.

The message was well understood, and no changes were suggested. Both Zanzibar and Mainland participants suggested adding more fruit such as pineapples, pawpaw, mangoes and avocados to the image. In Mainland, oranges were deemed difficult to access, so the suggestion was made to remove this fruit from the image.

In some instances, the message and image were incorrectly interpreted as “a variety of food”. *Ugali* is the most consumed staple on Mainland, and it was suggested to be depicted clearly in the image. Most participants could not identify the sweet potatoes in the image, and therefore, a variety of tubers were suggested for inclusion in the image. “Chapati” (a thin pancake of unleavened whole-meal bread cooked on a griddle) was removed from the original image to create space for the tubers.

The message and image were well understood; however, the need for education on the specific diseases mentioned were expressed. There was a general understanding that these foods are found in supermarkets and should be avoided. There was a recommendation to add biscuits, more hard-boiled sweets, and sugar sweetened drinks to the image.


**Maternal, Infant and Young Child Feeding (MIYCF) messages and images.**


The MIYCF messages were introduced with the following pre-amble statement: The first thousand days of a child’s life starts in pregnancy and continues until two years of age. This is a very important time to ensure a child grows well on food, love and care and becomes a productive adult.

There are three messages and images suggested for the MIYCF period.

The MIYCF messages were already in use and well understood and communicated widely, and no further testing was conducted for understanding of the messages. Suggested changes to the images were related to religious aspects of covering of pregnant and breastfeeding women’s faces and bodies for Zanzibar specifically. These changes were incorporated in the final set of images for Zanzibar (please see Section 3.3).

The message and image were well understood. However, in discussion of this message, some issues were raised in reference to available funds to afford healthy snacks. The suggestions made for changes to an accompanying image were as follows:
○Image with children around a table depicting peanuts and vegetables;○A child in school uniform being handed the suggested snacks by an elder or father;○A child standing next to the suggested snacks.

Since the image was already in the format of the latter suggestion, no further changes were made to this image.

A few participants raised concerns about the availability of money to buy healthy food in general, but also in reference to the message on supplying breakfast to children before school. It was mentioned that preparing breakfast is taxing on caregivers and that parents would need motivation to follow this message. This is due to limited time availability in the mornings.

Participants required clarity on what the woman was cooking in the image; therefore, the consistency of the porridge was changed to portray a “grittier” texture in the revised image. Another suggestion for change to the image included a mother providing tea and *chapati* to her children. It was also recommended that fruit be added to the image.

In Zanzibar, participants requested that the head and body of the women cooking should be covered. These changes were incorporated in the final set of images for Zanzibar (please see Section 3.3).

This message was generally not well understood by Mainland FGDs. Participants associated ‘school-age’ with younger children up to primary school rather than children of all ages attending school. It was felt that implementation would be difficult because most parents leave home early to go to work, but mostly because packing a lunchbox for children was unfamiliar as they receive porridge at school through the school nutrition programme. This was considered a better approach to reducing inequality and avoiding stigmatizing children whose parents may not be able to afford to pack a lunchbox.

“*The message should insist on lunch at school to avoid inequality at school which can create problems for the poor children*.”(Participant 1 MD2)

Children are not encouraged to bring lunch to school by teachers, and parents were uncertain that children would be able to take care of packed lunches. Participants were especially worried that giving their children packed food will lead to fights/bullying at school. Participants also discouraged giving money to buy food at school/tuck-shop.

“*I would draw a picture showing all the children having lunch at school and not packed lunch which may create conflicts*.”(Participant 3MI1)

Suggestions for changes to the image included:○A parent provides a child with a safe storage container containing food for school;○A child washing hands with a lunchbox depicted in close proximity;○Add a school building to the background;○Add a tap for washing hands;○Add a shared plate so that it represents a school feeding scheme.

A combination of porridge, depicting the school nutrition programme and shared lunchboxes were included in the revised image.

The message was well understood, but the image stimulated a lot of debate. The discussions were related to the man in the picture washing dishes. It was interpreted that if a man is washing the dishes/utensils, then he is not working and earning an income for the household. It was suggested that the man should be replaced by a woman, although some participants felt that the man could be depicted. Another suggestion was the inclusion of a pile of clean dishes in the image.

“*It will not be understood because men do not work in the kitchen. The woman should be in the picture instead and more food should be included*.”(Participant 6 MI1)

Participants also felt that the image looked too modern and proposed that it should show a typical Tanzanian environment including the inside and outside of the house where both men and women should be engaging in cleaning activities. Other suggested changes included depicting clean water, safely stored food, and sweeping of the floor.

“*The picture should show a clean house, with a clean kitchen, and the hair of the cook must be covered, and the other picture should show a house in a dirty environment and food covered with flies, for comparison.*”(Participant 3 MI1)

As all the other FBDG images feature women (e.g., preparing and cooking food), the research team felt that this was the most appropriate image to include a man sharing the workload of domestic tasks. The image was therefore reworked to show a man and a young boy collecting water from an outside tap.

This message was well understood, but the focus was rather on the hygiene aspect of clean water rather than making it the drink of choice daily. The message therefore needs to be accompanied with education on increasing intake of clean, safe water. A suggestion was made to indicate water boiling on a stove or water being purified, and/or a bucket showing that the water is clean/has been purified. In the revised image, activities around water collection and purification were portrayed. The man drinking bottled water was also replaced by a woman drinking water from a glass.

Participants articulated the meaning of both the message and the image very well. They commented that the image complemented the message well. A concern was raised that the government allows the trade in alcohol and cigarettes, hinting that the government and the proposed FBDG message are saying the opposite, which are two conflicting policy positions of government (health and trade policy).

Participants proposed there should be a person affected by the consumption of alcohol and cigarettes depicted in the image. The researchers discussed this suggestion and felt that the image should support the word “avoid” and should reflect a positive action of saying “no” to these products. No further changes were therefore made to the image.

This message was well understood, but quite a few suggestions were made to improve the acceptability of the image, including to show different kinds of activities, e.g., agriculture/gardening, a woman carrying water, skipping rope, a game such as “nage” (ball game) and swimming. Another suggestion was made to separate men and women while doing any activity and to cover women’s legs and arms, particularly by Zanzibar participants. In some of the Mainland FGDs, there was discomfort with women and the elderly engaging in physical activity. Therefore, this message should be accompanied by education about the benefits of physical activity for all age groups.

### 3.4. Final Considerations

Suggestions for changes to the proposed FBDG messages and images were incorporated as far as feasible.

During a pre-final online workshop in early 2021 between the researchers, TWG and FAO, the representatives from Zanzibar requested that all the applicable images be tweaked to portray Muslim dress specifications. It was eventually decided to prepare a separate set of images for Mainland and Zanzibar with the distinction of Muslim dress for all the images intended for Zanzibar.

The research team suggested that Message 1 (Everybody, young and old, should enjoy eating a variety of foods from different food groups every day to stay healthy and strong) can be used as an over-arching message for the FBDG messages, and Image 1 could be retained to depict a planetary healthy plate for Tanzania.

These suggestions were put forward to the TWG from Mainland and Zanzibar for their final consideration.

## 4. Discussion

This research study aimed to determine the understanding, acceptability, and feasibility of proposed FBDG messages and images amongst consumers in Tanzania. The importance of nutrition throughout the life cycle [28,29] combined with the value of focussing on food as the single strongest lever to optimize human health and environmental sustainability [30], inspired the researchers and the TWG to adopt a lifecycle approach with a focus on sustainable eating in the development of the proposed messages and images for the food-based dietary guidelines of the country.

It was evident from the findings of this study that in general, participants understood the meaning of the proposed FBDG messages and found them to be feasible, bearing in mind that a large proportion of the population live in poverty [9]. Furthermore, the images accompanying the messages were mostly well received and regarded as acceptable. Participants made practical suggestions for improved applicability and understanding, which was incorporated in as far as feasible to the final proposed set of FBDG messages and images for Tanzania. Since the images will be copied, sometimes in black and white, the graphic designer was mindful of the amount of detail that could be changed or added to certain images. Too much detail could distort the images when copies are made, and the focus of the image could be lost.

Although participants displayed some knowledge of dietary concepts, misinformation and lack of information on some aspects of nutrition were evident. In particular, the message on staple foods (Message 6: Eat staples such as cereals, starchy roots, tubers or plantains every day for a strong and active body.) was interpreted as “a variety of food” by some participants. This could be due to a diet based on staple foods and prevailing poor dietary diversity in the country [22]. These finding are consistent with widespread financial constraints, poor household food security and low socio-economic status in many Sub-Saharan countries, despite some degree of economic growth, and political and social transition [34]. The COVID-19 pandemic is expected to exacerbate undernutrition and food insecurity due to, among other factors, the vulnerability and weaknesses of already fragile food—and strained healthcare systems. Inequities in food and health systems worsen inequalities in nutrition outcomes that, in turn, can lead to more inequity, fuelling a vicious cycle [34]. It is therefore more important now than ever to scale-up efforts to support and educate the public about healthy and sustainable diets. In the context of the current study, the overarching message (Everybody, young and old, should enjoy eating a variety of foods from different food groups every day to stay healthy and strong) should be widely communicated and promoted, to ensure understanding of the staple message and image in particular.

An encouraging finding is that participants realised the need and expressed a desire for nutrition education to accompany the dissemination of the FBDG messages and images. It has been shown that social and behaviour change communication can effectively improve nutrition knowledge [35]. This could include interpersonal (counselling, education support groups), mass media (television, community radio, printed media, social and/or mobile technology) and community mobilization efforts (health days, campaigns) and corresponds with the suggestions made by participants for dissemination efforts.

Most studies on healthy diets and sustainable food production agree that a diet rich in plant-based foods with fewer animal source foods is good for human health and the environment [30]. A planetary healthy plate should consist of half a plate of vegetables and fruits and the other half should contain whole grains, plant protein sources, unsaturated plant oils, and (optionally) small amounts of animal protein sources. However, it is recognised that some populations depend on growing crops and raising livestock [30]. In addition, many populations suffer from high burdens of undernutrition and struggle to meet micronutrient needs from plant source foods alone. Evidence suggests that infants and young children also need animal source foods to grow and develop. Given these considerations, the role of animal source foods in people’s diets must be carefully considered in each context alongside local and regional realities [30]. It was therefore recommended that the “planetary healthy plate” image, field tested with Message 1, could be retained to convey information on sustainable eating to consumers in Tanzania.

According to a global review of FBDGs from 90 countries, the food groups most adhered to were the starchy staples (e.g., rice and potatoes) and fruit and vegetables [36]. Adequate intakes of fruit and vegetables are associated with a reduced risk of chronic diseases and body weight management. The WHO and FAO recommend that adults consume at least five servings of fruits and vegetables per day excluding starchy vegetables. Regardless of the evidence showing protective effects of fruits and vegetables, intakes are still inadequate in many countries, especially LMICs. For this reason, public health efforts and improved strategies, notably well-planned and behaviour-focused nutrition education intervention to promote fruit and vegetable intake, are critical [37]. Differences in names and cooking methods for certain foods on the Mainland and in Zanzibar, particularly for vegetables, made wording of the message challenging. For example, in Zanzibar, vegetables such as carrots, onions and tomatoes are referred to as spices. After much deliberation in the TWG, the vegetable message and image were finalized as follows: “Eat different coloured vegetables like tomatoes, onions, carrots, okra and green leafy vegetables every day to prevent and reduce risk of diseases”. The images were adapted to depict the most consumed, culturally acceptable and available products.

From the eight countries in Africa with published FBDGs (Benin, Kenya, Namibia, Nigeria, Seychelles, Sierra Leone, South Africa and Zambia), South Africa [27] and Zambia have a set of Paediatric FBDGs, and the Seychelles revised 2020 version has a section on IYCF. It was therefore deemed imperative that the FBDGs and images for Tanzania should include messages and visual illustrations to portray the importance of optimal nutrition during the first 1000 days of life [28,29]. According to the Tanzanian National Nutrition Survey (2018), 96.6% of children 0–23 months were ever breastfed and almost 58% of infants younger than six months were exclusively breastfed. This is an improvement from 2014 (41.1%). In Zanzibar, a significant increase in exclusive breastfeeding was observed between 2014 and 2018 (19.7% to 30.0%). Lower levels of exclusive breastfeeding on the Island are due, among other reasons, to religious practices at birth (e.g., smearing honey in the newborn’s mouth and giving water). At national level, it was reported that 86.8% of children aged 6 to 8 months were timeously introduced to complementary food, but dietary diversity remains sub-optimal [13]. A study by Mgongo et al. (2014) [38] revealed that although mothers understood the concept of exclusive breastfeeding and were positive about the practice, there were many barriers that mothers faced. Among the barriers were poor support and conflicting advice from influential people, including mothers-in-law, friends and healthcare workers. Returning to work and the pressures of earning an income was also highlighted as a barrier. Similar findings have been reported elsewhere on the continent, notably East-Africa (Tanzania, Ethiopia, Mozambique and Kenya) [39] and South Africa [40]. The Tanzanian government has invested in nutrition education at primary health care level, and it was clear from the pilot testing of the FBDG messages that IYCF messages have been widely communicated. These activities as well as more support for working mothers, should be strengthened and continued to further improve IYCF practices in the country.

Eating breakfast is recommended due to its association with improved macro- and micronutrient intakes, BMI and lifestyle. Breakfast is also widely promoted, particularly for school-aged children and adolescents, to improve cognitive function and academic performance [41]. School nutrition programmes aim to address short-term hunger and nutrient deficiencies, improve school attendance and performance, and support local agriculture and the economy [42]. School nutrition programmes further provide benefits for the physical, mental, and psychosocial development of school-age children and adolescents, particularly in LMICs [43]. In the current study, the proposed FBDG messages and images aimed at school-aged children elicited a lot of debate. Barriers that were mentioned to implementation of these messages included time constraints for parents to prepare breakfast and lunch packets/boxes, poverty, fear of bullying of children who take food to school and the fact that public schools in Tanzania have a nutrition programme in place. In the light of challenges worsened by the COVID-19 pandemic, governments have in some cases been forced to reduce social support, such as school nutrition programmes [34]. The most vulnerable and marginalised individuals and groups often rely on these support initiatives and there is a real risk that, as nations try to recover from the pandemic, the gains that were made pre-COVID in reducing hunger and malnutrition may be lost [34]. For these reasons, the promotion of eating breakfast and taking a lunchbox to school, if possible, remain important, as well as provision of school meals, particularly in LMICs.

### Limitations

Differences in names and cooking methods for certain foods on the Mainland and in Zanzibar made wording of the message and images challenging. Cultural differences, especially in clothing and dressing also made it difficult to have one representative image for a concept for the Republic of Tanzania (Mainland and Zanzibar). This was resolved by the final decision between the research team, TWG and FAO to prepare separate sets of FBDG messages and images with a few distinct differences in clothing/dressing and certain foods.

## 5. Conclusions and Recommendations

The proposed FBDGs messages and visual illustrations for Tanzania have been field-tested for understanding, acceptability and feasibility. The study results indicate a general awareness of the messages, but some rewording of certain messages and editing of certain images were suggested to facilitate the comprehension of the message and image. The TWG for Zanzibar FBDGs and Mainland FBDGs incorporated the messages and images in the respective FBDGs Technical Manuals that were compiled in the last quarter of 2021 and are due to be launched in 2022. It is recommended that the field-tested FBDG messages and images with the incorporated suggested changes be adopted to form part of the national nutrition education efforts by the Tanzanian government.

Once adopted, the FBDGs messages and images should be used to educate various stakeholders, including parents, caregivers, healthcare providers and educators on appropriate nutritional practices for children and adults. The messages can be used to guide implementation of nutrition education to promote healthy diets while simultaneously addressing the abundance of misinformation on food and nutrition. The use of FBDGs can ensure consistent messages to support the healthy growth and development of young children in Tanzania as well as general nutritional well-being of adult consumers in the country.

It is also strongly recommended that FBDGs should be applied more broadly and not just within food and nutrition education programmes. Experts propose a food-systems approach whereby FBDGs should be incorporated into programmes, policies and other publications of national and sub-national departments in sectors such as agriculture, social development and safety and security in order to contribute to the achievement of sustainable healthy diets. Once implemented, the use and impact of the FBDGs messages should be monitored and evaluated [44].

## Figures and Tables

**Figure 1 nutrients-14-02705-f001:**
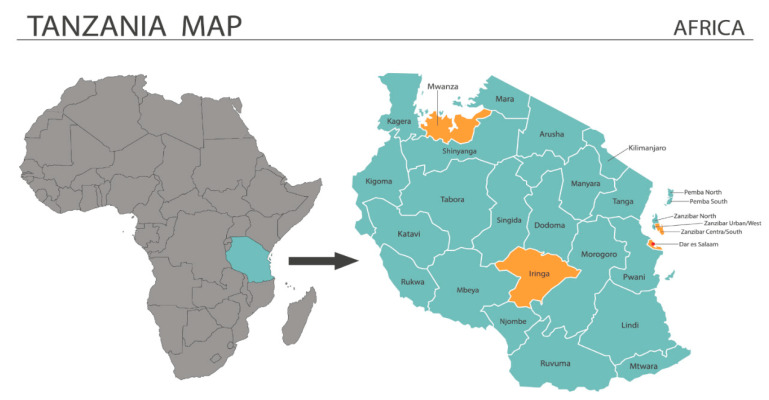
Map of Africa indicating the Republic of Tanzania and country-level districts. Source: https://stock.adobe.com/images (accessed on 26 April 2022).

**Figure 2 nutrients-14-02705-f002:**
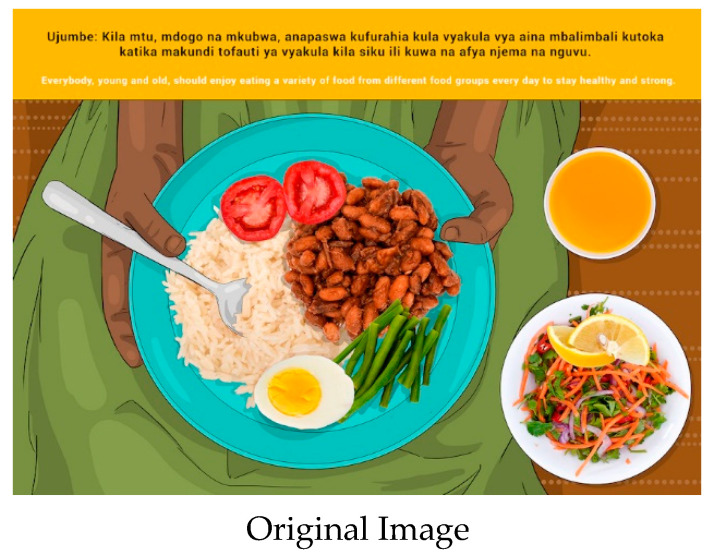
Everybody, young and old, should enjoy eating a variety of foods from different food groups every day to stay healthy and strong.

**Figure 3 nutrients-14-02705-f003:**
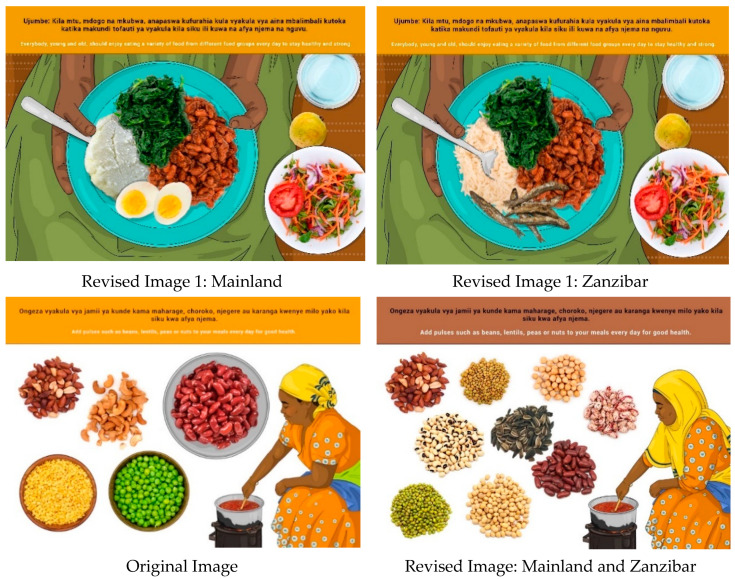
Add pulses such as beans, lentils, peas or nuts to your meals every day for good health.

**Figure 4 nutrients-14-02705-f004:**
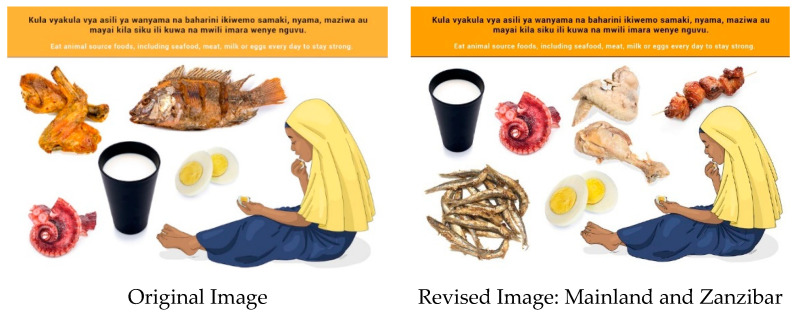
Eat animal source foods, including seafood, meat, milk or eggs every day to stay strong.

**Figure 5 nutrients-14-02705-f005:**
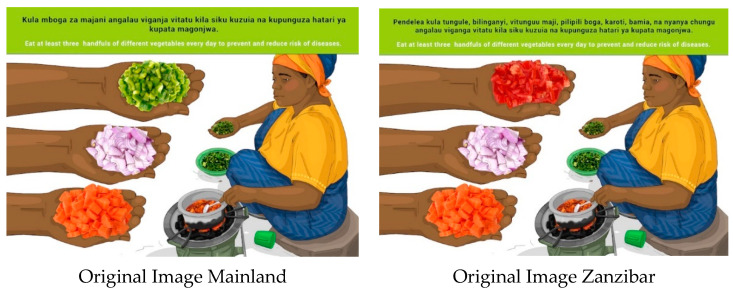
Eat different vegetables, at least three handfuls every day to prevent and reduce risk of diseases.

**Figure 6 nutrients-14-02705-f006:**
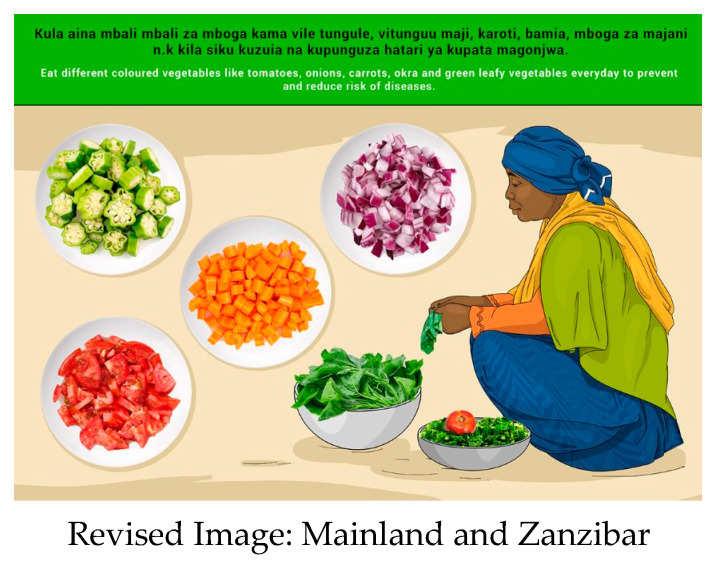
Revised message and image 4 for Mainland and Zanzibar: Eat different coloured vegetables such as tomatoes, onions, carrots, okra and green leafy vegetables every day to prevent and reduce risk of diseases.

**Figure 7 nutrients-14-02705-f007:**
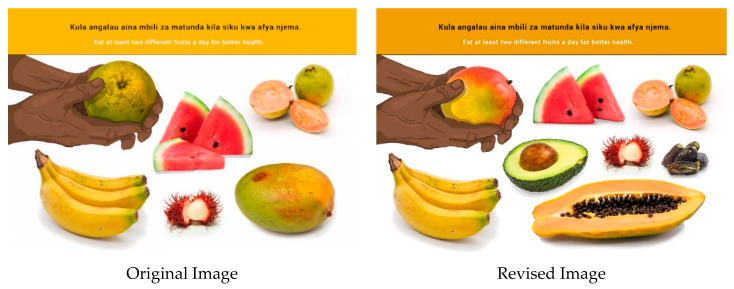
Eat at least two types of fruits every day for better health.

**Figure 8 nutrients-14-02705-f008:**
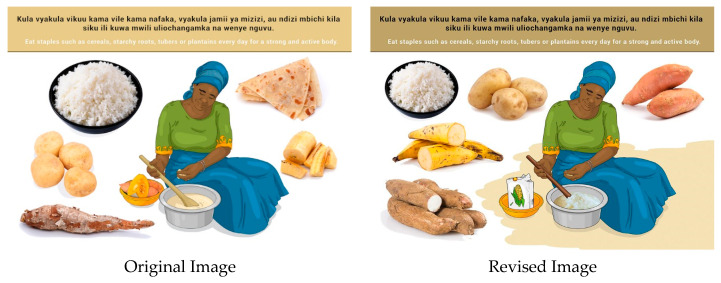
Eat staples such as cereals, starchy roots, tubers or plantains every day for a strong and active body.

**Figure 9 nutrients-14-02705-f009:**
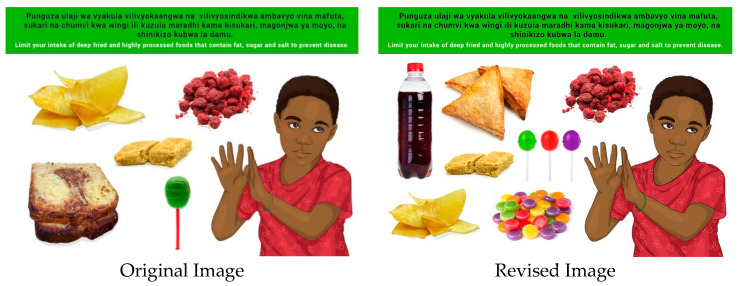
Limit your intake of deep fried and highly processed foods that contain fat, sugar and salt to prevent disease such as high blood pressure, diabetes and heart diseases.

**Figure 10 nutrients-14-02705-f010:**
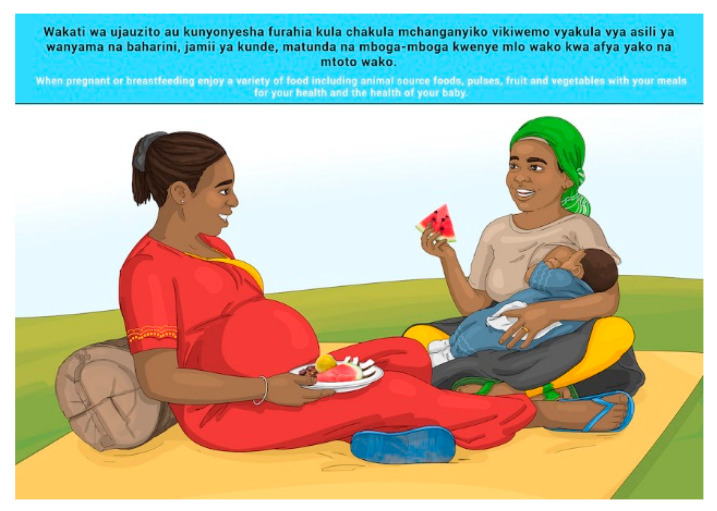
When pregnant or breastfeeding enjoy a variety of food including animal source foods, pulses, fruit and vegetables with your meals for your health and the health of your baby.

**Figure 11 nutrients-14-02705-f011:**
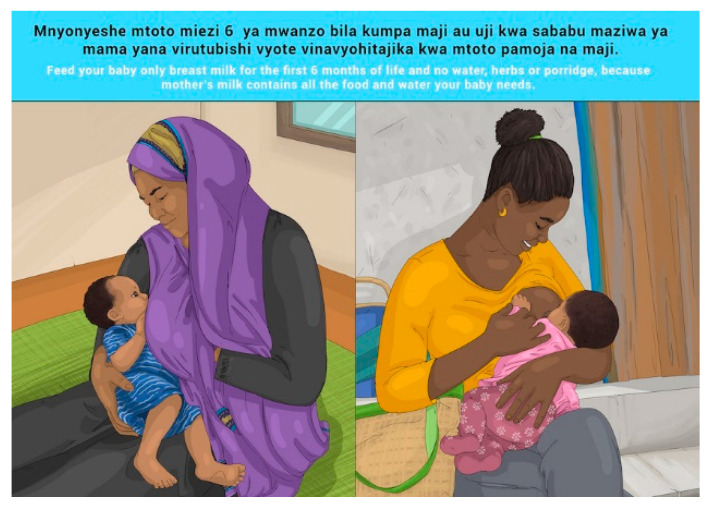
Feed your baby only breast milk for the first 6 months of life and no water, herbs or porridge, because mother’s milk contains all the food and water your baby needs.

**Figure 12 nutrients-14-02705-f012:**
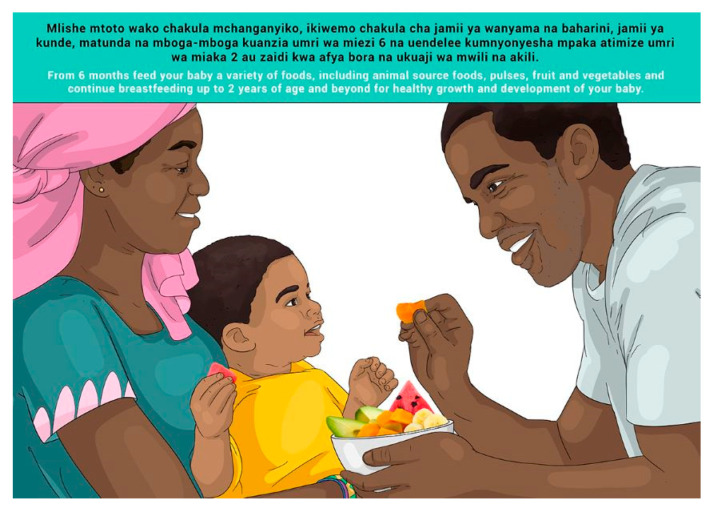
From 6 months feed your baby a variety of foods, including animal source foods, pulses, fruit and vegetables and continue breastfeeding up to 2 years of age and beyond for healthy growth and development of your baby.

**Figure 13 nutrients-14-02705-f013:**
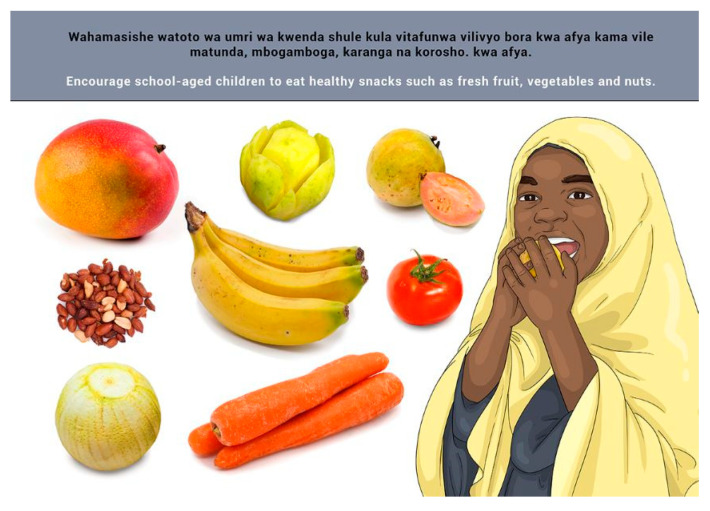
Encourage school-aged children to eat healthy snacks such as fresh fruit, vegetables and nuts.

**Figure 14 nutrients-14-02705-f014:**
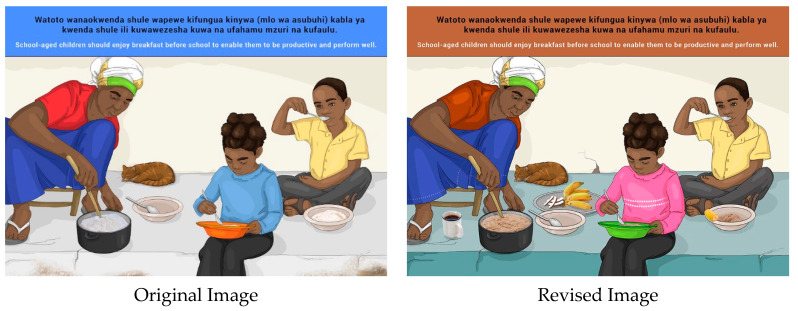
School-aged children should enjoy breakfast before school to enable them to be productive and perform well.

**Figure 15 nutrients-14-02705-f015:**
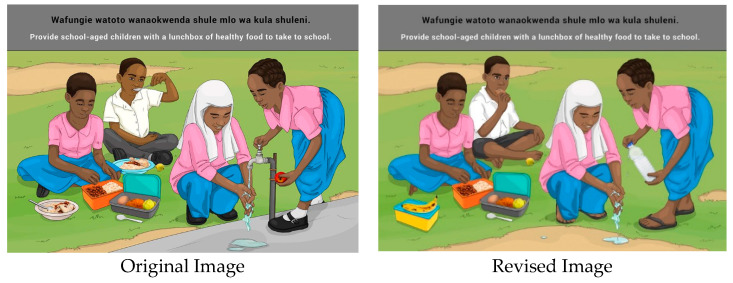
Provide school-aged children with a lunchbox to take to school.

**Figure 16 nutrients-14-02705-f016:**
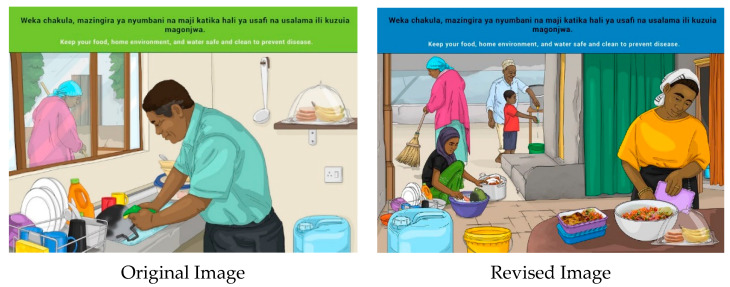
Keep your food, home environment and water safe and clean to prevent diseases.

**Figure 17 nutrients-14-02705-f017:**
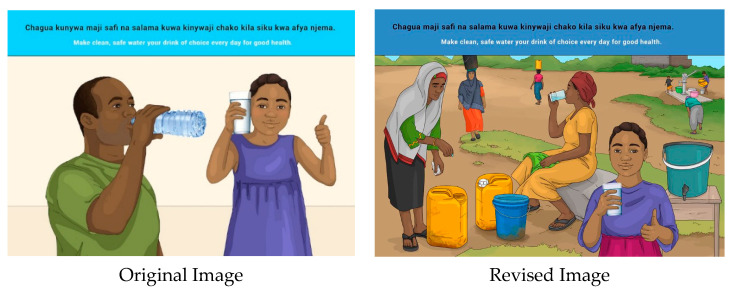
Make clean, safe water your drink of choice every day for good health.

**Figure 18 nutrients-14-02705-f018:**
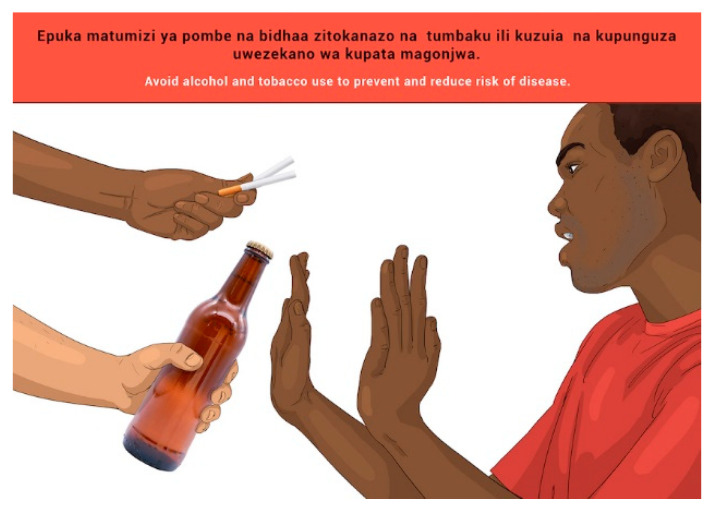
Avoid drinking alcohol and tobacco use to prevent and reduce risk of diseases.

**Figure 19 nutrients-14-02705-f019:**
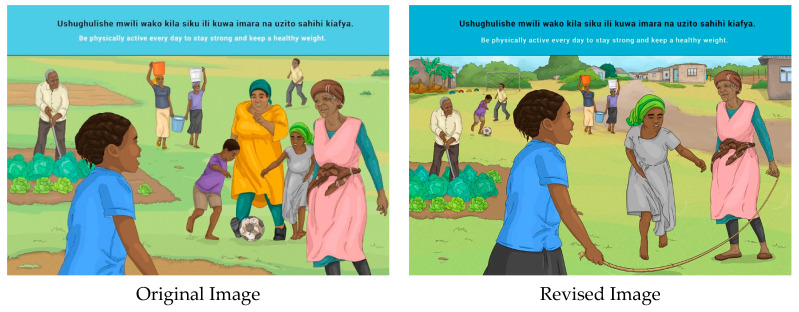
Be physically active every day to stay strong and keep a healthy weight.

**Table 1 nutrients-14-02705-t001:** DRAFT FBDG messages for field-testing with the population of Tanzania.

Message Nr	E = English S = Kiswahili	Message Wording
1	E	Everybody, young and old, should enjoy eating a variety of foods from different food groups every day to stay healthy and strong.
1	S	*Ujumbe: Kila mtu, mdogo na mkubwa, anapaswa kufurahia kula vyakula vya aina mbalimbali kutoka katika makundi tofauti ya vyakula kila siku ili kuwa na afya njema na nguvu.*
2	E	Add pulses such as beans, lentils, peas or nuts to your meals every day for good health.
2	S	*Ongeza vyakula vya jamii ya kunde kama maharage, choroko, njegere au karanga kwenye milo yako kila siku kwa afya njema.*
3	E	Eat animal source foods, including seafood, meat, milk or eggs every day to stay strong.
3	S	*Kula vyakula vya asili ya wanyama na baharini ikiwemo samaki, nyama, maziwa au mayai kila siku ili kuwa na mwili imara wenye nguvu.*
4	E	MAINLAND: Eat different vegetables, at least three handfuls every day to prevent and reduce risk of diseases.
		ZANZIBAR: Enjoy eating tomatoes, eggplant, onions, sweet pepper, carrot, okra and bitter tomatoes at least three handfuls every day to prevent and reduce risk of diseases.
4	S	MAINLAND: *Kula mboga za majani angalau viganja vitatu kila siku kuzuia na kupunguza hatari ya kupata magonjwa.*
		ZANZIBAR: Pendelea kula tungule, bilinganyi, vitunguu maji, pilipili boga, karoti, bamia, na nyanya chungu angalau viganga vitatu kila siku kuzuia na kupunguza hatari ya kupata magonjwa.
5	E	Eat at least two types of fruits every day for better health.
5	S	*Kula angalau aina mbili za matunda kila siku kwa afya njema.*
6	E	Eat staples such as cereals, starchy roots, tubers or plantains every day for a strong and active body.
6	S	*Kula chakula kikuu kama vile nafaka, vyakula vya mizizi (viazi, mihogo, nk) au ndizi za kupika kila siku kuwa na mwili imara na wenye nguvu.*
7	E	Limit your intake of deep fried and highly processed foods that contain fat, sugar and salt to prevent disease such as high blood pressure, diabetes and heart diseases.
7	S	*Punguza ulaji wa vyakula vilivyokaangwa na vilivyosindikwa* *ambavyo vina mafuta, sukari na chumvi kwa wingi ili kuzuia maradhi kama kisukari, magonjwa ya moyo, na shinikizo kubwa la damu.*
8	E	Statement: The first thousand days of a child’s life starts in pregnancy and continues until two years of age. This is a very important time to ensure a child grows well on food, love and care and becomes a productive adult.
8	S	*Tamko: Siku elfu moja za mwanzo wa maisha ya mtoto huanzia wakati mama amepata ujauzito na kuendelea mpaka mtoto anapotimiza umri wa miaka miwili. Muda huu ni wa muhimu kuhakikisha mtoto anapewa chakula vizuri, kwa upendo na kujaliwa ili aweze kukua vizuri na kuwa mtu mzima mwenye nguvu na afya.*
8	E	When pregnant or breastfeeding enjoy a variety of food including animal source foods, pulses, fruit and vegetables with your meals for your health and the health of your baby.
8	S	*Wakati wa ujauzito au kunyonyesha furahia kula chakula mchanganyiko* *vikiwemo vyakula vya asili ya wanyama na baharini, jamii ya kunde, matunda na mboga-mboga kwenye mlo wako kwa afya yako na mtoto wako.*
9	E	Feed your baby only breast milk for the first 6 months of life and no water, herbs or porridge, because mother’s milk contains all the food and water your baby needs.
9	S	*Mnyonyeshe mtoto miezi 6 ya mwanzo bila kumpa maji au uji kwa sababu maziwa ya mama yana virutubishi vyote vinavyohitajika kwa mtoto pamoja na maji.*
10	E	From 6 months feed your baby a variety of foods, including animal source foods, pulses, fruit and vegetables and continue breastfeeding up to 2 years of age and beyond for healthy growth and development of your baby.
10	S	*Mlishe mtoto wako chakula mchanganyiko, ikiwemo chakula cha jamii ya wanyama na baharini, jamii ya kunde, matunda na mboga-mboga kuanzia umri wa miezi 6 na uendelee kumnyonyesha mpaka atimize umri wa miaka 2 au zaidi kwa afya bora na ukuaji wa mwili na akili.*
11a	E	Encourage school-aged children to eat healthy snacks such as fresh fruit, vegetables and nuts.
11a	S	*Wahamasishe watoto wa umri wa kwenda shule kula vitafunwa (asusa) vilivyo bora kwa afya kama vile matunda, mbogamboga, karanga na korosho.*
11b	E	School-aged children should enjoy breakfast before school to enable them to be productive and perform well.
11b	S	*Watoto wanaokwenda shule wapewe kifungua kinywa (mlo wa asubuhi) kabla ya kwenda shule ili kuwawezesha kuwa na ufahamu mzuri na kufaulu.*
11c		Provide school-aged children with a lunchbox to take to school.
11c		*Wafungie watoto wanaokwenda shule mlo wa kula shuleni.*
12	E	Keep your food, home environment and water safe and clean to prevent diseases.
12	S	*Weka chakula, mazingira ya nyumbani na maji katika hali ya usafi na usalama ili kuzuia magonjwa.*
13	E	Make clean, safe water your drink of choice every day for good health.
13	S	*Chagua kunywa maji safi na salama kuwa kinywaji chako kila siku kwa afya njema.*
14	E	Avoid drinking alcohol and tobacco use to prevent and reduce risk of diseases.
14	S	*Epuka matumizi ya pombe na bidhaa zitokanazo na tumbaku ili kuzuia na kupunguza uwezekano wa kupata magonjwa.*
15	E	Be physically active every day to stay strong and keep a healthy weight.
15	S	*Ushughulishe mwili wako kila siku ili kuwa imara na uzito sahihi kiafya.*

**Table 2 nutrients-14-02705-t002:** Sociodemographic characteristics of participants.

Variable		n (%) ^α^
District/region	Dar es Salaam	96 (29.6)
	Mwanza	98 (30.5)
	Iringa	72 (22.2)
	Kusini (Zanzibar)	58 (17.9)
Age	* Average age in years (Mean ± SD)	35.01 ± 11.62
Women with children	Yes	277 (85.5)
	No	47 (14.5)
Number of children	* Average number of children (mean ± SD)	3.22 ± 2.30
Ages of children	* Average age of children in years (mean ± SD)	13.93 ± 10.48
Mother/caregiver of children under 5 years of age	Yes	174 (53.7)
	No	150 (46.3)
Home Language	Kiswahili	322 (99.4)
	* missing data	2 (0.6)
Education status	None	15 (4.6)
	Primary school/Grade 1–7	196 (60.5)
	Secondary school/Grade 8–10 (Form i–iv)	107 (33.0)
	Secondary school/Grade 11–12 (Form v–vi)	2 (0.6)
	Diploma	2 (0.6)
	Tertiary education (college, university)	0 (0.0)
	* missing data	2 (0.85)
Employment status	Employed	193 (59.6)
	Unemployed	76 (23.5)
	Unemployed, not looking for work	55 (17.0)
Role relating to food in the household	Provide or contribute money for food	248 (76.5)
	Decide what food should be bought or used in the house	240 (74.1)
	Purchase food	252 (77.8)
	Prepare food	302 (93.2)
	Grow food for use in household	212 (65.4)
	Grow food and sell produce for money	158 (48.8)
	Other	17 (5.2)

^α^ Percentage calculated from total sample N = 324. * Averages given in mean and standard deviations (mean ± SD).

## Data Availability

Data supporting reported results are available to view on reasonable request.

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
