# Peer review of "Development and Field-Testing of Proposed Food-Based Dietary Guideline Messages and Images amongst Consumers in Tanzania"

_nutrients, 2022, doi:10.3390/nu14132705_

Round 1

Reviewer 1 Report

Thank you for the opportunity to review a well-prepared and written article by experts in the field - with the importance of the topic and the need to implement it.

With some minor comments for your consideration:

- does the bioethics committee's consent be needed for focus research - because the participants were also interviewed (on their characteristics)?

- the correctness of the combination of the terms “malnutrition” and “over-nutrition” should be considered, or it should be explained that the authors write about over-nutrition with a deficiency of nutrients “Similar to other LMICs, Tanzania suffers a triple burden of malnutrition (i.e., under-nutrition, micronutrient deficiencies and over-nutrition)”, because we define malnutrition as lack of proper nutrition, caused by not having enough to eat, not eating enough of the right things, or being unable to use the food that one does eat (lines 99-100)

- the first sentence of the discussion should be a response to the implementation of the research objective; the first paragraph of the discussion repeats the content of the introduction

Overall, the article is very interesting and provides very relevant information.

Author Response

Dear Reviewers. Thank you very much for your constructive feedback and comments. Please see attached file containing your comments and how the authors addressed it. Kind regards. 

Reviewer 2 Report

It is a nice piece of work and attractive to read, although it is a long manuscript.

My comments are:

- regarding the inclusion process; could you describe in more detail how their inclusion process/ purposive sampling led to participants that cover the wide range of diversity as described in the introduction (e.g. literacy, religion)?

- it would be helpful to have some more information about the topic guide that was used for the focus group. How was the guide developed?

- Analyses: could you describe in more detail how transcripts were coded? E.g.: Did you use inductive of deductive coding, or both?

- there seems to be a typing error in line 466: it should say image 4 instead of 3?

Author Response

(The authors gave the same response as above.)
